# Towards a Fairer Non-negative Matrix Factorization

## Abstract

Topic modeling, or more broadly, dimensionality reduction, techniques provide powerful tools for uncovering patterns in large datasets and are widely applied across various domains. We investigate how Non-negative Matrix Factorization (NMF) can introduce bias in the representation of data groups, such as those defined by demographics or protected attributes. We present an approach, called Fairer-NMF, that seeks to minimize the maximum reconstruction loss for different groups relative to their size and intrinsic complexity. Further, we present two algorithms for solving this problem. The first is an alternating minimization (AM) scheme and the second is a multiplicative updates (MU) scheme which demonstrates a reduced computational time compared to AM while still achieving similar performance. Lastly, we present numerical experiments on synthetic and real datasets to evaluate the overall performance and trade-offs of Fairer-NMF.

## 1 Introduction

Machine Learning (ML) and Artificial Intelligence (AI) have seen a huge surge in uses and applications in recent times and are being used in nearly every aspect of society. Despite this, there are serious and critical issues that propagate bias, disseminate unfair outcomes, and affect racial and social justice (CBS News, 2023; Ongweso, 2019; Truong, 2020). This lack of equity typically stems from many sources, from bias in the data to algorithmic bias and even post-processing decisions (Barocas et al., 2023; Mehrabi et al., 2021). In this work, we focus on inequities stemming from both the lack of fair representation of the data as well as algorithmic bias in the treatment of that data. For instance, ML techniques are often used in medical applications to inform diagnostics, make medical predictions, and assess health risks. At the heart of these applications lies the need for an algorithm to uncover hidden themes or trends, that either explain some studied medical phenomenon or are used downstream for a learning task like classification or prediction. When the data used is skewed toward certain populations (typically by uneven representation), this can lead to misclassification and poorly explained patterns for minority populations. This includes populations defined by attributes such as gender, race, ethnicity, and other classifying factors.

In this work, we focus on addressing this problem for topic modeling, or more generally, dimensionality reduction tasks. We specifically consider Non-negative Matrix Factorization (NMF), a powerful method used in a wide array of ML applications. It serves as a mathematically tangible example of a method for discovering data trends that highlight the inequities we wish to study (Lee & Seung, 1999; 2001). Indeed, at the heart of NMF is a seemingly simple objective function that asks that the factorization has a small *average error* which is typical of objective functions used in ML. Because such objectives ask only for overall low error, information extracted from an imbalanced dataset tends to emphasize traits or patterns within the majority group but gives less attention to groups that are less represented in the data. This can be very concerning in many applications; for example in medical applications, patterns that fit the majority group might not be true for the minority group. Additionally, post-algorithmic decisions are often made from these outputs without knowledge of how well-represented each individual is.

This work aims to explore a fairer alternative objective function for NMF under a specific framework of fairness and present algorithmic implementations for solving this formulation. Before one can seek to promote fairness, it is essential to first define what fairness means. Defining fairness and fighting against bias and discrimination has existed long before the advent of machine learning (Saxena, 2019). Indeed, definitions

of fairness vary across tasks and settings and are generally non-universal which contributes to the difficulty of solving such problems (Barocas et al., 2023; Mehrabi et al., 2021). We emphasize here, as we do in the title, that our goal must be humble; we seek a *fairer—not fair*—formulation, and even that will only be fairer for certain contexts and applications. Nonetheless, we view this as an important step forward, which will hopefully lead to ML algorithms with more transparency and flexibility for the end user to identify and mitigate bias. In our numerical experiments, we discuss the trade-offs and overall performance of our approach.

## 1.1 Contribution

In this work, we showcase on synthetic and real datasets how standard NMF provides a low-dimensional representation that explains groups within the data in an often inequitable way. These inequities may be catastrophic in many applications. We propose and study an alternative formulation of NMF, which we call Fairer-NMF, that seeks to minimize the maximum reconstruction loss across all groups. We discuss why in many applications this approach may lead to more fair outcomes across the entire population and cases where such fairness criterion is no longer "fair". Furthermore, we present two algorithms for solving this problem; namely, an alternating minimization (AM) scheme and a multiplicative scheme (MU) scheme. We discuss the motivations behind these approaches, their properties, and numerical performance. Last, we showcase experimentally how Fairer-NMF performs relative to standard NMF on several datasets, and conclude with a discussion about the critical need for fairer machine learning methods, as well as an understanding that fairness is very application-dependent and nuanced.

## 1.2 Organization

In Section 2, we present related works in fair unsupervised learning techniques, particularly dimensionality reduction. We provide an overview of NMF, its applications, and existing algorithms in Section 3. In Section 4, we further discuss the objective function of Standard NMF and define the fairness criterion of our proposed NMF formulation called Fairer-NMF. Then, in Section 5, we present two algorithms for solving Fairer-NMF. Lastly, in Section 6, we present numerical experiments on synthetic and real data to demonstrate the performance of Fairer-NMF.

## 2 Related Works

In this section, we focus on related work that is most pertinent to our study, rather than providing an exhaustive review.

### 2.1 Fair Unsupervised Learning

The topic of fairness in clustering has recently gained significant interest in the machine learning community with Chierichetti et al. (2017) leading the first work on fair clustering. Due to the difficulty in defining and enforcing fairness criteria for unsupervised learning tasks, including clustering techniques, many different fairness notions for clustering exist (Backurs et al., 2019; Chen et al., 2019; Ghadiri et al., 2021; Mahabadi & Vakilian, 2020). An overview of fair clustering is given in Chhabra et al. (2021). Fairness issues in recommender systems have also recently attracted increasing attention, leading to the emergence of works aimed at mitigating bias (e.g., Li et al. (2021); Zhu et al. (2020)). In Wang et al. (2023), the authors provide a survey on the fairness of recommender systems. Some works have also proposed fairness-aware matrix factorizations for recommender systems (e.g., Togashi & Abe (2022)) including federated approaches (e.g., Liu et al. (2022)).

### 2.2 Fair Principal Component Analysis

Dimensionality reduction is one of the most common tasks employed in data science. Principal component analysis (PCA) is a fundamental and widely used dimensionality reduction algorithm. In Samadi et al. (2018), the authors investigate how PCA might inadvertently introduce bias. The analysis is performed on

two real datasets *Labeled Faces in the Wild* (Huang et al., 2008) and *Default of Credit Card Clients* (Yeh & Lien, 2009). The numerical experiments show that PCA incurs much higher average reconstruction error for one population than another (e.g., lower- versus higher-educated individuals), even when the populations are of similar sizes. The authors established a formulation called, *Fair PCA*, that addresses this bias under a specific framework of fairness. For a given matrix $\mathbf{Y} \in \mathbb{R}^{a \times n}$ denote by $\hat{\mathbf{Y}} \in \mathbb{R}^{a \times n}$ the optimal rank-$d$ approximation of $\mathbf{Y}$. Given $\mathbf{Z} \in \mathbb{R}^{a \times n}$ with rank at most $d$, the reconstruction loss is defined as,

$$loss(\mathbf{Y}, \mathbf{Z}) := \|\mathbf{Y} - \mathbf{Z}\|_F^2 - \|\mathbf{Y} - \hat{\mathbf{Y}}\|_F^2.$$

Consider a data matrix $\begin{bmatrix} \mathbf{A} \\ \mathbf{B} \end{bmatrix} \in \mathbb{R}^{m \times n}$ where the rows in $\mathbf{A}$ and $\mathbf{B}$ are samples in the data belonging to a group $A$ and $B$, respectively. The problem of finding a projection into $d$-dimensions in *Fair PCA* is defined as solving,

$$\min_{\mathbf{U} \in \mathbb{R}^{m \times n}, \text{rank}(\mathbf{U}) \leq d} \max \left\{ \frac{1}{|A|} loss(\mathbf{A}, \mathbf{U}_A), \frac{1}{|B|} loss(\mathbf{B}, \mathbf{U}_B) \right\} \tag{1}$$

where $\mathbf{U}_A$ and $\mathbf{U}_B$ are the matrices with rows in $\mathbf{U}$ corresponding to groups $A$ and $B$, respectively. The authors propose a polynomial-time algorithm for solving this problem that involves solving a semidefinite program to find nearly-optimal low-dimensional representations. In Tantipongpipat et al. (2019), the authors introduce a multi-criteria dimensionality reduction problem, where multiple objectives are optimized simultaneously. One application of this model is capturing several fairness criteria in dimensionality reduction, such as the Fair PCA problem proposed in Samadi et al. (2018).

Our work is motivated by Samadi et al. (2018). Indeed, the criterion of Fair PCA, Equation (1), minimizes the maximum of the average reconstruction loss across different groups. Similar to Samadi et al. (2018), socially fair k-means clustering introduced in Ghadiri et al. (2021) seeks to minimize the maximum of the average clustering cost across different groups. In Ghadiri et al. (2021), the term *socially fair* is introduced to refer to the min-max approach for clustering which has since been adapted more generally to unsupervised learning tasks.

### 2.3 Fair Generalized Low-rank Models

In Buet-Golfouse & Utyagulov (2022), the authors discuss NMF within the broad framework of generalized low-rank models (GLRM) and investigate the fairness of GLRMs in unsupervised learning settings. In particular, the considered NMF problem is in the form of an unconstrained optimization problem where non-negativity constraints are added as penalty terms to the standard cost function. This is distinct from our approach where we consider a constrained optimization problem for NMF leading to enforced non-negativity in the learning and outcome of the model.

Buet-Golfouse & Utyagulov (2022) further presents a broad fair framework for GLRMs with several examples of models and fairness criteria. The authors consider fairness criteria that aim to reduce the disparity across group-wise average costs. In particular, a penalized learning approach is considered where a disparity penalty term is added to the cost function of the model. The disparity is defined as the difference between a groups average loss and the overall loss (a constant term across all groups). This is in contrast to our approach where we consider the difference between a groups average loss and the overall loss *restricted* to the group.

The authors employ algorithms that utilize the biconvex nature of these problems to solve them. Various fairness criteria for the penalty term are discussed; however, the authors primarily analyze the weighted Log-Sum Exponential (wLSE) function defined as:

$$T(z_1, \cdots, z_K) = \frac{1}{\alpha} \log \left( \sum_{k=1}^{K} w_k e^{\alpha z_k} \right)$$

where $\alpha > 0, w_k \geq 0$ for all $k$, $\sum_{k=1}^{K} w_k = 1$, and $z_k$ is the average cost of a group $k$. For $\alpha = \infty$, wLSE recovers the max function. The authors discuss how the hyper-parameter $\alpha$ enables one to "interpolate" between the traditional GLRM problem and its "fair" min-max formulation and why one may wish to choose $\alpha \neq \infty$. In the numerical experiments, the min-max problem is approximated by minimizing $T$ with $\alpha = 10^5$

and the trade-offs between average loss and group disparity is studied by varying values of $\alpha$. Additionally, the experiments explore the generalization performance of different GLRMs on out-of-sample data across various datasets.

In this paper, we seek to solve the min-max fairness problem on the constrained optimization problem for NMF with a different objective function from Buet-Golfouse & Utyagulov (2022). We detail this approach in Section 4. Further, our analysis seeks to understand and investigate the performance of Fairer-NMF compared to standard NMF overall and on a group level.

## 3 Background

### 3.1 Notation

We use boldfaced upper-case Latin letters (e.g., $\mathbf{A}$) to denote matrices and $\mathbf{A} \in \mathbb{R}_{\geq 0}^{m \times n}$ to denote an $m \times n$ matrix with real non-negative entries. The Frobenius norm of a matrix $\mathbf{X}$ is denoted by $\|\mathbf{X}\|$.

For a dataset partitioned into two (or more) mutually exclusive sample groups, we write the data matrix $\mathbf{X}$ with $m$ number of samples and $n$ number of features in block format as,

$$\mathbf{X} = \begin{bmatrix} \mathbf{X}_A \\ \mathbf{X}_B \end{bmatrix} \in \mathbb{R}_{\geq 0}^{m \times n} \quad \text{where } \mathbf{X}_A \in \mathbb{R}_{\geq 0}^{m_1 \times n}, \mathbf{X}_B \in \mathbb{R}_{\geq 0}^{m_2 \times n}.$$

The matrices $\mathbf{X}_A$ and $\mathbf{X}_B$ are the matrices with rows in $\mathbf{X}$ corresponding to group $A$ and $B$, respectively. In general, we write $|A|$ to denote the size of group $A$. Here, $|A| = m_1$ and $|B| = m_2$ and $m_1 + m_2 = m$. Additionally, the notation $\mathbf{X}_A$ generally means we restrict to rows of $\mathbf{X}$ with sample indices given by $A$.

The notation $\mathbf{A}/\mathbf{B}$ indicates entrywise division, $\mathbf{A} \odot \mathbf{B}$ entrywise multiplication, and $\mathbf{AB}$ standard matrix multiplication. The notation $\mathbf{e}_\ell \in \mathbb{R}^L$ indicates the standard basis vector in $\mathbb{R}^L$ where $\ell$-th entry in the vector $\mathbf{e}$ is 1 and all other entries are zero.

### 3.2 Non-negative Matrix Factorization (NMF)

*Topic modeling* is a machine learning technique used to reveal latent themes or patterns from large datasets. A popular technique for topic modeling that provides a low-rank approximation of a matrix is non-negative matrix factorization (NMF) (Lee & Seung, 1999; 2001). Given a non-negative matrix $\mathbf{X} \in \mathbb{R}_{\geq 0}^{m \times n}$ and a target dimension $r \in \mathbb{N}$, NMF decomposes $\mathbf{X}$ into a product of two low-dimensional non-negative matrices: $\mathbf{W} \in \mathbb{R}_{\geq 0}^{m \times r}$ and $\mathbf{H} \in \mathbb{R}_{\geq 0}^{r \times n}$, such that

$$\mathbf{X} \approx \mathbf{WH}$$

We consider $\mathbf{X}$ to be a data matrix where the rows represent data samples and the columns represent data features. Typically, $r > 0$ is chosen such that $r \ll \min\{m, n\}$ to reduce the dimension of the original data matrix or reveal hidden patterns in the data. The matrix $\mathbf{W}$ is called the *representation matrix* and $\mathbf{H}$ is called the *dictionary matrix*. The rows of $\mathbf{H}$ are generally referred to as *topics*, which are characterized by features of the dataset. Each row of $\mathbf{W}$ provides the approximate representation of the respective row in $\mathbf{X}$ in the lower-dimensional space spanned by the rows of $\mathbf{H}$. Thus, the data points are well approximated by an additive linear combination of the topics.

We note that in the NMF literature, $r$ is referred to as the target dimension, the number of desired topics, or the desired non-negative rank. It is a user-specified hyperparameter and can be estimated heuristically. The *non-negative rank* of a matrix $\mathbf{X}$ is the smallest integer $r^* > 0$ such that there exists an exact NMF decomposition: $\mathbf{X} = \mathbf{WH}$ where $\mathbf{W} \in \mathbb{R}_{\geq 0}^{m \times r^*}$ and $\mathbf{H} \in \mathbb{R}_{\geq 0}^{r^* \times n}$. Computing the exact non-negative rank of a matrix is NP-hard (Vavasis, 2010). Therefore, several formulations for the non-negative approximation, $\mathbf{X} \approx \mathbf{WH} \in \mathbb{R}_{\geq 0}^{m \times n}$, have been studied (Cichocki et al., 2009; Lee & Seung, 1999; 2001) that seek to minimize the reconstruction error of the decomposition.

**Definition 3.1 (Relative Reconstruction Error)** *Suppose* $\mathbf{X} \in \mathbb{R}_{\geq 0}^{m \times n}$ *and* $r < \min\{m, n\} \in \mathbb{N}$. *For a given* $\mathbf{W} \in \mathbb{R}_{\geq 0}^{m \times r}$ *and* $\mathbf{H} \in \mathbb{R}_{\geq 0}^{r \times n}$ *we define the reconstruction error of* $\mathbf{X}$ *as* $\|\mathbf{X} - \mathbf{WH}\|$ *and the relative reconstruction error of* $\mathbf{X}$ *as* $\|\overline{\mathbf{X}} - \mathbf{WH}\|/\|\mathbf{X}\|$.

One of the most popular formulations of finding an NMF approximation uses the Frobenius norm as a measure of the reconstruction error,

$$\underset{\mathbf{W}\in\mathbb{R}_{\geq 0}^{m\times r},\mathbf{H}\in\mathbb{R}_{\geq 0}^{r\times n}}{\operatorname{argmin}} \|\mathbf{X} - \mathbf{W}\mathbf{H}\|^2. \tag{2}$$

Throughout the paper, we refer to this formulation as *rank-r NMF* or *standard NMF with rank r*. For simplicity and as common in the literature of NMF, we will refer to non-negative rank simply as rank.

Many numerical optimization techniques can be applied to find local minima for the NMF problem defined in Equation 2 (e.g., Cichocki et al. (2009); Kim et al. (2008); Kim & Park (2008); Lin (2007); Paatero & Tapper (1994)). Note that although (2) is a non-convex optimization problem, it is convex in $\mathbf{W}$ when $\mathbf{H}$ is held fixed and vice-versa. Thus, an *alternating minimization* (AM) approach (see e.g., Bertsekas (1997)) can be used to find local minima:

$$\mathbf{W}^{(k)} \leftarrow \underset{\mathbf{W}\in\mathbb{R}_{\geq 0}^{m\times r}}{\operatorname{argmin}} \|\mathbf{X} - \mathbf{W}\mathbf{H}^{(k)}\|^2$$

$$\mathbf{H}^{(k)} \leftarrow \underset{\mathbf{H}\in\mathbb{R}_{\geq 0}^{r\times n}}{\operatorname{argmin}} \|\mathbf{X} - \mathbf{W}^{(k)}\mathbf{H}\|^2$$

where $k$ denotes the $k$-th iteration. Both of these convex problems are non-negative least squares problems, and specialized solvers exist to find solutions.

Another minimization method is the *multiplicative updates* (MU) method proposed in Lee & Seung (2001). The method can be viewed as an entrywise projected gradient descent algorithm. The choice of stepsize for each entry of the updating matrix results in multiplicative (rather than additive) update rules that ensure nonnegativity. The algorithm performs alternating steps in updating $\mathbf{W}$ and $\mathbf{H}$:

$$\mathbf{W} \leftarrow \mathbf{W} \odot \frac{\mathbf{X}\mathbf{H}^\top}{\mathbf{W}\mathbf{H}\mathbf{H}^\top}$$

$$\mathbf{H} \leftarrow \mathbf{H} \odot \frac{\mathbf{W}^\top\mathbf{X}}{\mathbf{W}^\top\mathbf{W}\mathbf{H}}$$

The multiplicative updates algorithm is commonly used due to its ease of implementation, the absence of the need for user-defined hyperparameters, and desirable monotonicity properties (Lee & Seung, 2001).

NMF has garnered increasing attention due to its effectiveness in handling large-scale data across various domains. In image processing, NMF is employed for tasks like feature extraction and perceptual hashing (Rajapakse et al., 2004; Tang et al., 2013; Lee & Seung, 1999). In the field of text mining, it has proven useful for document clustering and semantic analysis (Berry & Browne, 2005; Xu et al., 2003). In the medical field, it has been employed on applications ranging from fraud detection (Zhu et al., 2011) and phenotyping (Joshi et al., 2016) to studying trends in health and disease from record or survey data (Hamamoto et al., 2022; Hassaine et al., 2020; Johnson et al., 2024; Vendrow et al., 2020). Indeed, due to the nonnegativity constraints, NMF acquires a parts-based, sparse representation of the data (Lee & Seung, 1999). When the features are naturally non-negative, this approach often enhances interpretability compared to traditional methods like Principal Components Analysis (PCA) (Lee & Seung, 1999).

## 4 Fairer-NMF Formulation

In this section, we highlight some characteristics of NMF on a group level and propose our approach, Fairer-NMF.

### 4.1 Standard NMF Objective

Suppose a dataset consists of two mutually exclusive groups $A$ and $B$. For example, these groups could be divided based on a protected attribute in the data. Suppose $A$ has size $|A| = m_1$ and $B$ has size $|B| = m_2$.

We can write the data matrix $\mathbf{X}$ with $m$ number of samples and $n$ number of features as:

$$\mathbf{X} = \begin{bmatrix} \mathbf{X}_A \\ \mathbf{X}_B \end{bmatrix} \in \mathbb{R}_{\geq 0}^{m \times n} \quad \text{where } \mathbf{X}_A \in \mathbb{R}_{\geq 0}^{m_1 \times n}, \mathbf{X}_B \in \mathbb{R}_{\geq 0}^{m_2 \times n}.$$

Indeed, $\mathbf{X}_A$ and $\mathbf{X}_B$ are the matrices with rows in $\mathbf{X}$ corresponding to groups $A$ and $B$, respectively.

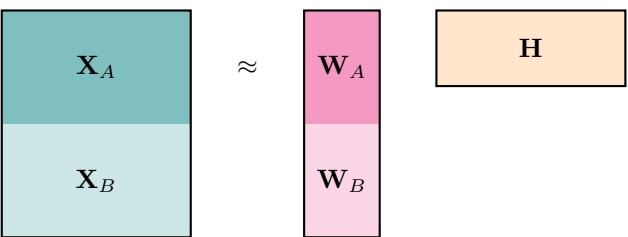

Figure 1: Illustration of NMF applied to a data matrix $\mathbf{X}$ that consists of two submatrices $\mathbf{X}_A$ and $\mathbf{X}_B$. The matrices $\mathbf{W}_A$ and $\mathbf{W}_B$ are the representation matrices corresponding to $\mathbf{X}_A$ and $\mathbf{X}_B$, respectively and $\mathbf{H}$ the common dictionary matrix.

We can write the standard NMF of $\mathbf{X} \approx \mathbf{WH}$ as,

$$\begin{bmatrix} \mathbf{X}_A \\ \mathbf{X}_B \end{bmatrix} \approx \begin{bmatrix} \mathbf{W}_A \\ \mathbf{W}_B \end{bmatrix} \mathbf{H} \tag{3}$$

where $\mathbf{W}_A \in \mathbb{R}_{\geq 0}^{m_1 \times r}$ is the representation matrix corresponding to $\mathbf{X}_A$, $\mathbf{W}_B \in \mathbb{R}_{\geq 0}^{m_2 \times r}$ is the representation matrix corresponding to $\mathbf{X}_B$, and $\mathbf{H} \in \mathbb{R}_{\geq 0}^{r \times n}$ is the common dictionary matrix. An illustration of the decomposition is given in Figure 1. In the standard NMF, defined in Equation (2), we have:

$$\operatorname*{argmin}_{\mathbf{W} \in \mathbb{R}_{\geq 0}^{m \times r}, \mathbf{H} \in \mathbb{R}_{\geq 0}^{r \times n}} \|\mathbf{X} - \mathbf{WH}\|^2 = \operatorname*{argmin}_{\substack{\mathbf{W}_A \in \mathbb{R}_{\geq 0}^{m_1 \times r}, \mathbf{W}_B \in \mathbb{R}_{\geq 0}^{m_2 \times r} \\ \mathbf{H} \in \mathbb{R}_{\geq 0}^{r \times n}}} \left( \|\mathbf{X}_A - \mathbf{W}_A\mathbf{H}\|^2 + \|\mathbf{X}_B - \mathbf{W}_B\mathbf{H}\|^2 \right).$$

Note that both groups are weighted equally and we seek to minimize the sum of the reconstruction error of each group in the joint decomposition. The problem above can be written for $L$ mutually exclusive groups,

$$\operatorname*{argmin}_{\mathbf{W} \in \mathbb{R}_{\geq 0}^{m \times r}, \mathbf{H} \in \mathbb{R}_{\geq 0}^{r \times n}} \|\mathbf{X} - \mathbf{WH}\|^2 = \operatorname*{argmin}_{\substack{\mathbf{W}_\ell \in \mathbb{R}_{\geq 0}^{m_\ell \times r}, \forall \ell \in \{1, \cdots, L\} \\ \mathbf{H} \in \mathbb{R}_{\geq 0}^{r \times n}}} \sum_{\ell=1}^{L} \|\mathbf{X}_\ell - \mathbf{W}_\ell\mathbf{H}\|^2. \tag{4}$$

This standard objective function is designed to perform well *on average*. It seeks an overall low reconstruction error which disregards the size and complexity of each data group. For example, in the case of an imbalanced dataset where $|A| \gg |B|$ an overall low reconstruction error does not guarantee that the reconstruction error restricted to group $B$ is low. Additionally, the standard objective function does not pay attention to the complexity of the data groups.

Figure 2 shows the relative average reconstruction error in percent for three groups in a randomly generated synthetic dataset, both the average and standard deviation over several trials. The full details of these experiments are described in Section 6.2. NMF is performed both on the full dataset and on each group separately. We observe that the reconstruction generated by standard NMF on the full data is generally best able to explain the large group, despite the medium group having a similar complexity.

Minimizing the maximum *reconstruction error* may seem desirable, but this approach favors optimizing for the group with inherently higher complexity when the group sizes and magnitudes are all the same. Similarly, this approach favors optimizing for the group with the larger size and magnitude when the rank is the same for all groups. In Section 4.2, we present a fairness criterion that takes into account the size and complexity of the data groups.

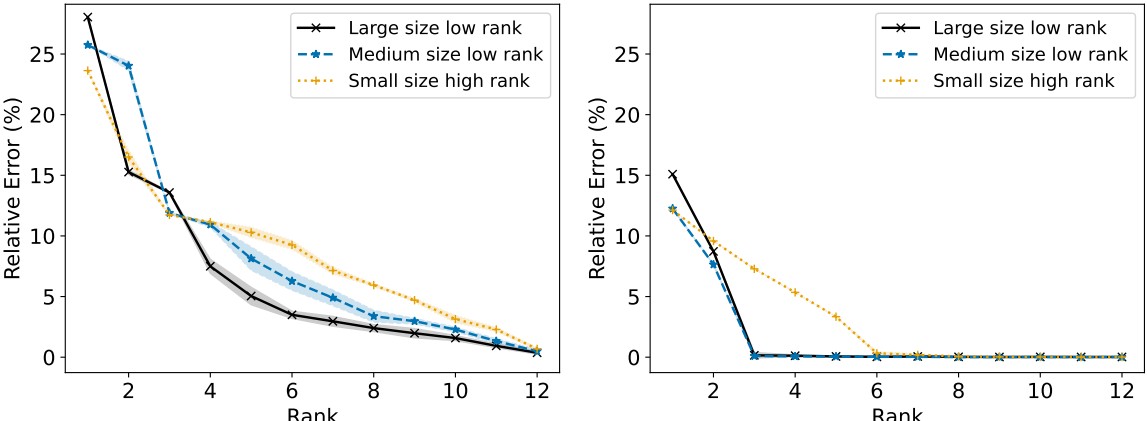

Figure 2: The Relative Error (%) of each group is reported. Left: Standard NMF applied on the entire synthetic data matrix. Right: Standard NMF applied on the synthetic data matrix of each group individually.

## 4.2 Fairer-NMF Objective

In our consideration of a fairer NMF, we work under a min-max fairness framework with a criterion similar to that of Fair-PCA (Samadi et al., 2018). We consider the following definition of *reconstruction loss* of a group.

**Definition 4.1 (Relative Reconstruction Loss)** *Suppose we have a group data matrix* $\mathbf{X}_\ell \in \mathbb{R}_{\geq 0}^{m_\ell \times n}$, *desired rank* $r \in \mathbb{N}$, *and* $E_\ell$ *representing the error obtained by replacing* $X_\ell$ *with a* $r$-*rank NMF approximation. For a given* $\mathbf{W} \in \mathbb{R}_{\geq 0}^{m \times r}$ *and* $\mathbf{H} \in \mathbb{R}_{\geq 0}^{r \times n}$, *we define the relative reconstruction loss of* $\mathbf{X}_\ell$ *as,*

$$\frac{\|\mathbf{X}_\ell - \mathbf{W}\mathbf{H}\| - E_\ell}{\|\mathbf{X}_\ell\|}.$$

**Remark 4.2** *A reasonable choice for* $E_\ell$ *is to take*

$$E_\ell = \mathbb{E}_{\mathbf{X}_\ell^*} \|\mathbf{X}_\ell - \mathbf{X}_\ell^*\|$$

*where the expectation is taken over the* $\mathbf{X}_\ell^*$ *obtained from a specific randomized implementation of rank-*$r$ *NMF on* $\mathbf{X}_\ell$.

We provide details on the numerical approximation in Section 5. We again highlight that finding an optimal non-negative rank-$r$ NMF of a matrix $\mathbf{X}_\ell \in \mathbb{R}_{\geq 0}^{m_\ell \times n}$ is NP-hard (Vavasis, 2010). Taking the expectation over a randomized NMF implementation allows $E_\ell$ to compensate both for the underlying dimensionality of the group and for how difficult it is for the NMF algorithm to find a good representation for the group.

Roughly speaking, the reconstruction loss is the difference between how well group $\mathbf{X}_\ell$ is reconstructed via a standard NMF model trained on the entire data $\mathbf{X}$ and a standard NMF model trained on $\mathbf{X}_\ell$ only. We further note that we normalize by the Frobenius norm of the group matrix to mitigate for the size of the group and the actual varying magnitudes of $\|\mathbf{X}_\ell\|$.

We seek to minimize the maximum of the average reconstruction loss across $L$ different groups:

$$\min_{\mathbf{W} \in \mathbb{R}_{\geq 0}^{m \times r} \mathbf{H} \in \mathbb{R}_{\geq 0}^{r \times n}} \quad \max_{\ell \in \{1, \cdots, L\}} \left\{ \frac{\|\mathbf{X}_\ell - \mathbf{W}_\ell \mathbf{H}\| - E_\ell}{\|\mathbf{X}_\ell\|} \right\} \tag{5}$$

First, we note that the second term in the objective function $E_\ell$ is a fixed pre-computed constant for a given group $\ell$ as defined in Remark 4.2. This optimization problem seeks to learn a common NMF model that

minimizes the maximum deviation from the group-wise "optimal" NMF approximation across all groups. We emphasize that our goal is to learn a common non-negative low-rank NMF model for all groups, rather than separate NMF models for each group.

Unfortunately, the loss function (5) is incomplete. We wish to select $\mathbf{H}$ that minimizes the loss:

$$\max_{\ell \in \{1, \cdots, L\}} \left\{ \frac{\|\mathbf{X}_\ell - \mathbf{W}_\ell \mathbf{H}\| - E_\ell}{\|\mathbf{X}_\ell\|} \right\}.$$

However, in general, $\mathbf{W}$ is under-specified. We can freely perturb $\mathbf{W}_\ell$ for any group that does not attain the maximum loss and achieves another solution of (5), provided the loss for group $\ell$ does not grow too large. This is because the group representation matrices $\mathbf{W}_\ell$ all act independently of each other. To resolve this issue, we choose

$$\min_{\mathbf{W}_\ell \in \mathbb{R}_{\geq 0}^{m_\ell \times r}} \|\mathbf{X}_\ell - \mathbf{W}_\ell \mathbf{H}\|$$

for all $\ell$ independently of each other, which is equivalent to choosing

$$\min_{\mathbf{W} \in \mathbb{R}_{\geq 0}^{m \times r}} \|\mathbf{X} - \mathbf{W}\mathbf{H}\|. \tag{6}$$

We remark the optimal $\mathbf{W}$ for a fixed $\mathbf{H}$ may give different losses for different groups even if $\mathbf{H}$ is the exact minimum given by (5). While this may result in one group having a lower loss than another, this inequality is "free" in the sense that it does not come at the expense of another group. Therefore, in Fairer-NMF we seek to minimize both (5) and (6), prioritizing the first over the second.

## 5 Algorithms

We present two algorithms for solving the Fairer-NMF problem formulation.

### 5.1 Estimating $E_\ell$

As discussed in Remark 4.2, for a group $\ell$, a reasonable choice for the estimate of the optimal rank-$r$ error, $E_\ell$, is to take the expectation over the error obtained from a specific randomized implementation of rank-$r$ NMF for the group. This leads to a natural algorithm for estimating $E_\ell$. It suffices to sample the single group NMF reconstruction $T$ times and take the average. This is described in Algorithm 1.

---

**Algorithm 1:** Estimating $E_\ell$

**Input:** data matrix $\mathbf{X}_\ell \in \mathbb{R}_{\geq 0}^{m_\ell \times n}$ of group $\ell$; desired dimension $r \in \mathbb{N}$; number of estimates $T \geq 1$

1   $E \leftarrow 0$
2   **for** $t \leftarrow 1$ **to** $T$ **do**
3     $\mathbf{W}_\ell, \mathbf{H}_\ell \leftarrow \underset{\mathbf{W} \in \mathbb{R}_{\geq 0}^{m_\ell \times r}, \mathbf{H} \in \mathbb{R}_{\geq 0}^{r \times n}}{\operatorname{argmin}} \|\mathbf{X}_\ell - \mathbf{W}\mathbf{H}\|$       // Approximate using standard NMF
4     $E \leftarrow E + \|X_\ell - \mathbf{W}_\ell \mathbf{H}_\ell\|$
5   $E_\ell \leftarrow E/T$

**Output:** rank-$r$ error $E_\ell$

---

### 5.2 Alternating Minimization (AM) Scheme

The optimization problem (5) is non-convex with respect to $\mathbf{H}$ and $\mathbf{W}_\ell$ for all $\ell$. However, it is convex with respect to one of the factor matrices while all others are held fixed. Further, the corresponding constraint sets are convex. This allows us to solve the problem using an AM approach on a multi-convex problem as outlined in Algorithm 2. The AM scheme solves a convex problem in each minimization step, ensuring that there is a global minimum. Solving for $\mathbf{H}^{(k)}$ and $\mathbf{W}^{(k)}$ in Algorithm 2 is similar to solving a standard NMF problem using the AM approach.

---

**Algorithm 2:** Fairer-NMF: Alternating Minimization Scheme

**Input:** data matrix $\mathbf{X} = \begin{bmatrix} \mathbf{X}_1 \in \mathbb{R}_{\geq 0}^{m_1 \times n} \\ \vdots \\ \mathbf{X}_L \in \mathbb{R}_{\geq 0}^{m_L \times n} \end{bmatrix}$ with $L$ groups; desired dimension $r \in \mathbb{N}$

**1** Compute $E_\ell$ for each group $\ell$             `// e.g., through Algorithm 1`

**2** Randomly initialize $\mathbf{W}^{(0)} = \begin{bmatrix} \mathbf{W}_1^{(0)} \in \mathbb{R}_{\geq 0}^{m_1 \times r} \\ \vdots \\ \mathbf{W}_L^{(0)} \in \mathbb{R}_{\geq 0}^{m_L \times r} \end{bmatrix}$

**3** $k \leftarrow 0$

**4** **while** *not converged* **do**

**5**     $k \leftarrow k + 1$

**6**     $\mathbf{H}^{(k)} \leftarrow \underset{\mathbf{H} \in \mathbb{R}_{\geq 0}^{r \times d}}{\operatorname{argmin}} \ \underset{\ell \in \{1,\dots,L\}}{\max} \ \dfrac{\|\mathbf{X}_\ell - \mathbf{W}_\ell^{(k-1)}\mathbf{H}\| - E_\ell}{\|\mathbf{X}_\ell\|}$

**7**     $\mathbf{W}^{(k)} = \begin{bmatrix} \mathbf{W}_1^{(k)} \\ \vdots \\ \mathbf{W}_L^{(k)} \end{bmatrix} \leftarrow \underset{\mathbf{W} \in \mathbb{R}_{\geq 0}^{n \times r}}{\operatorname{argmin}} \|\mathbf{X} - \mathbf{W}\mathbf{H}^{(k)}\|$

**Output:** coefficient matrix $\mathbf{W}^{(k)} = \begin{bmatrix} \mathbf{W}_1^{(k)} \in \mathbb{R}_{\geq 0}^{m_1 \times r} \\ \vdots \\ \mathbf{W}_L^{(k)} \in \mathbb{R}_{\geq 0}^{m_L \times r} \end{bmatrix}$; dictionary matrix $\mathbf{H}^{(k)} \in \mathbb{R}_{\geq 0}^{r \times n}$

---

As remarked in Equation (4), the update rule of $\mathbf{W}^{(k)}$ in Algorithm 2 is equivalent to updating the representation matrix of each group $\ell$ as,

$$\mathbf{W}_\ell^{(k)} = \underset{\mathbf{W}_\ell \in \mathbb{R}_{\geq 0}^{n \times r}}{\operatorname{argmin}} \|\mathbf{X}_\ell - \mathbf{W}_\ell \mathbf{H}^{(k)}\|.$$

Consider the function $f$ defined as,

$$f(\mathbf{W}^{(k)}, \mathbf{H}^{(k)}) = \max_{\ell \in \{1,\dots,L\}} \frac{\|\mathbf{X}_\ell - \mathbf{W}_\ell^{(k)}\mathbf{H}^{(k)}\| - E_\ell}{\|\mathbf{X}_\ell\|}, \tag{7}$$

where $\mathbf{H}^{(k)}$ and $\mathbf{W}^{(k)}$ defined in Algorithm 2. Indeed,

$$\mathbf{H}^{(k)} \leftarrow \underset{\mathbf{H} \in \mathbb{R}_{\geq 0}^{r \times d}}{\operatorname{argmin}} f(\mathbf{W}^{(k)}, \mathbf{H})$$

is a convex optimization problem. Then, we have that the loss function $f$ is non-increasing $f(\mathbf{W}^{(k)}, \mathbf{H}^{(k)}) \leq f(\mathbf{W}^{(k-1)}, \mathbf{H}^{(k-1)})$ where equality is achieved at a stationary point. Thus, by iterating the updates of $\mathbf{H}^{(k)}$ and $\mathbf{W}^{(k)}$, we obtain a sequence of estimates whose loss values converge.

The two optimization functions in Algorithm 2 both fall under restricted classes of convex programs that admit specialized solvers. The problem

$$\min_{\mathbf{H} \in \mathbb{R}_{\geq 0}^{r \times d}} \ \max_{\ell \in \{1,\dots,L\}} \frac{\|\mathbf{X}_\ell - \mathbf{W}_\ell^{(k-1)}\mathbf{H}\| - E_\ell}{\|\mathbf{X}_\ell\|}$$

is equivalent to

$$\min_{\substack{\mathbf{H} \in \mathbb{R}_{\geq 0}^{r \times d} \\ t \in \mathbb{R}}} \ t$$

$$\text{subj. to} \quad \|\mathbf{X}_\ell - \mathbf{W}_\ell^{(k-1)}\mathbf{H}\| \leq t\|\mathbf{X}_\ell\| + E_\ell \qquad \forall \ell \in \{1,\dots,L\}$$

which is a second-order cone program (SOCP). The minimization problem for $W$ is equivalent to

$$\min_{\mathbf{W}\in\mathbb{R}_{\geq 0}^{n\times r}} \|\mathbf{X}-\mathbf{W}\mathbf{H}^{(k)}\|^2$$

which is a non-negative least squares (NNLS) problem, a specific type of quadratic program (QP).

## 5.3 Multiplicative Update (MU) Scheme

In addition to the AM scheme, we also adapt multiplicative update (MU) rules for the Fairer-NMF problem formulation. Consider an equivalent form to the loss function (7):

$$f(\mathbf{W}^{(k)}, \mathbf{H}^{(k)}) = \max_{\substack{\mathbf{c}\in\mathbb{R}^L \\ \|\mathbf{c}\|_1=1}} \underbrace{\sum_{\ell=1}^{L} \mathbf{c}_\ell \frac{\|\mathbf{X}_\ell - \mathbf{W}_\ell^{(k)}\mathbf{H}^{(k)}\| - E_\ell}{\|\mathbf{X}_\ell\|}}_{:=g(\mathbf{W}^{(k)}, \mathbf{H}^{(k)}, \mathbf{c})}.$$

Let $\mathbf{c}^{(k)}$ be our estimate of the maximizer $\mathbf{c}$ and consider an alternating approach by maximizing $g(\mathbf{W}, \mathbf{H}, \mathbf{c})$ in $\mathbf{c}$ and minimizing in $\mathbf{W}$ and $\mathbf{H}$. The maximizer $\mathbf{c}^{(k)}$ at the $k$-th iteration is simply $\mathbf{e}_{\ell_*}$ (the $\ell_*$-th standard basis vector) where,

$$\ell_* = \operatorname*{argmax}_{\ell\in\{1,\dots,L\}} \frac{\|\mathbf{X}_\ell - \mathbf{W}_\ell^{(k-1)}\mathbf{H}^{(k-1)}\| - E_\ell}{\|\mathbf{X}_\ell\|}.$$

Start with $\mathbf{c}^{(0)} = \mathbf{0}$. By setting $\mathbf{c}^{(k)} = \frac{k-1}{k}\mathbf{c}^{(k-1)} + \frac{1}{k}\mathbf{e}_{\ell_*}$ we update with a decreasing step size while ensuring $\|\mathbf{c}^{(k)}\|_1 = 1$. This is desirable as just setting $\mathbf{c}^{(k)} = \mathbf{e}_{\ell_*}$. Otherwise, using a fixed step size can result in too much oscillation in the largest loss group and therefore poor convergence.

For fixed $\mathbf{c}^{(k)}$ and $\mathbf{W}^{(k-1)}$, we minimize $g$ in $\mathbf{H}$ by selecting

$$\operatorname*{argmin}_{\mathbf{H}\in\mathbb{R}_{\geq 0}^{r\times d}} \sum_{\ell=1}^{L} \mathbf{c}_\ell^{(k)} \frac{\|\mathbf{X}_\ell - \mathbf{W}_\ell^{(k-1)}\mathbf{H}\|}{\|\mathbf{X}_\ell\|}. \tag{8}$$

A related but easier problem to solve is

$$\operatorname*{argmin}_{\mathbf{H}\in\mathbb{R}_{\geq 0}^{r\times d}} \sum_{\ell=1}^{L} \left(\mathbf{c}_\ell^{(k)}\right)^2 \frac{\|\mathbf{X}_\ell - \mathbf{W}_\ell^{(k-1)}\mathbf{H}\|^2}{\|\mathbf{X}_\ell\|^2} = \operatorname*{argmin}_{\mathbf{H}\in\mathbb{R}_{\geq 0}^{r\times d}} \|\tilde{\mathbf{X}} - \tilde{\mathbf{W}}\mathbf{H}\|^2 \tag{9}$$

for the following two block matrices:

$$\tilde{\mathbf{X}} = \begin{bmatrix} \mathbf{c}_1^{(k)}\mathbf{X}_1/\|\mathbf{X}_1\| \in \mathbb{R}_{\geq 0}^{m_1\times n} \\ \vdots \\ \mathbf{c}_L^{(k)}\mathbf{X}_L/\|\mathbf{X}_L\| \in \mathbb{R}_{\geq 0}^{m_L\times n} \end{bmatrix}, \tilde{\mathbf{W}} = \begin{bmatrix} \mathbf{c}_1^{(k)}\mathbf{W}_1^{(k-1)}/\|\mathbf{X}_1\| \in \mathbb{R}_{\geq 0}^{m_1\times r} \\ \vdots \\ \mathbf{c}_L^{(k)}\mathbf{W}_L^{(k-1)}/\|\mathbf{X}_L\| \in \mathbb{R}_{\geq 0}^{m_L\times r} \end{bmatrix}.$$

The solution of (9) attains a value of $g(\mathbf{W}^{(k)}, \mathbf{H}, \mathbf{c}^{(k)})$ that is at most $\sqrt{L}$ that of (8) that can be considered as a good choice of minimizer. Thus, we can select $\mathbf{H}^{(k)}$ according to the standard multiplicative update for $\tilde{\mathbf{X}}$ with $\tilde{\mathbf{W}}$. Once we've obtained $\mathbf{H}^{(k)}$, we can use the multiplicative update for the single group NMF problem $\mathbf{X}_\ell \approx \mathbf{W}_\ell\mathbf{H}$ to obtain $\mathbf{W}_\ell^{(k)}$ for each group $\ell$. This procedure is described in Algorithm 3.

We make three concluding remarks about this algorithm:

1. Enforcing $\|\mathbf{c}^{(k)}\|_1 = 1$ is actually unnecessary, as the multiplicative update is invariant under scalar multiplication for $\tilde{\mathbf{X}}$ and $\tilde{\mathbf{W}}$. Therefore it suffices to just set $\mathbf{c}^{(k)} = \mathbf{c}^{(k-1)} + \mathbf{e}_{\ell_*}$.

2. This algorithm is self-correcting (see below).

3. While a decaying update for $\mathbf{c}$ is empirically important for convergence, the exact rate chosen was selected for the simplicity of its implementation. It is likely that other decaying update rules would yield similarly good performance.

Let us consider the claim that this algorithm is self-correcting. Consider the vector $\mathbf{v} \in \mathbb{R}^L$ defined by

$$\mathbf{v}_\ell = \mathbf{c}_\ell^{(k)} \frac{\|\mathbf{X}_\ell - \mathbf{W}_\ell^{(k-1)}\mathbf{H}\|}{\|\mathbf{X}_\ell\|}.$$

When we replace (8) with (9) we move from minimizing the $\ell_1$ norm of $\mathbf{v}$ to minimizing the $\ell_2$ norm of $\mathbf{v}$. These are similar when $\mathbf{v}$ has most of its mass in a single component, which occurs when $\mathbf{c}^{(k)}$ has one coordinate that is much larger than the rest. Suppose an iteration results in a poor minimizer for $g$, for example because (8) and (9) have very different solutions. This is likely to manifest as one group having much higher loss than the others after this iteration. If this happens consistently over many iterations, the corresponding coordinate of $\mathbf{c}$ will increase faster. This results in (8) and (9) having similar solutions.

---

**Algorithm 3:** Fairer-NMF: Multiplicative Update Scheme

**Input:** data matrix $\mathbf{X} = \begin{bmatrix} \mathbf{X}_1 \in \mathbb{R}_{\geq 0}^{m_1 \times n} \\ \vdots \\ \mathbf{X}_L \in \mathbb{R}_{\geq 0}^{m_L \times n} \end{bmatrix}$ with $L$ groups; desired dimension $r \in \mathbb{N}$

**1** Compute $E_\ell$ for each group $\ell$         `// e.g., through Algorithm 1`

**2** Randomly initialize $\mathbf{H}^{(0)} \in \mathbb{R}_{\geq 0}^{r \times n}$, $\mathbf{W}^{(0)} = \begin{bmatrix} \mathbf{W}_1^{(0)} \in \mathbb{R}_{\geq 0}^{m_1 \times r} \\ \vdots \\ \mathbf{W}_L^{(0)} \in \mathbb{R}_{\geq 0}^{m_L \times r} \end{bmatrix}$

**3** Initialize $\mathbf{c}^{(0)} = \mathbf{0} \in \mathbb{R}^L$

**4** $k \leftarrow 0$

**5** **while** *not converged* **do**

**6**    $k \leftarrow k+1$

**7**    $\ell_* \leftarrow \underset{\ell \in \{1,\dots,L\}}{\mathrm{argmax}} \dfrac{\|\mathbf{X}_\ell - \mathbf{W}_\ell^{(k-1)}\mathbf{H}^{(k-1)}\| - E_\ell}{\|\mathbf{X}_\ell\|}$

**8**    $\mathbf{c}^{(k)} \leftarrow \mathbf{c}^{(k-1)} + \mathbf{e}_{\ell_*}$         `// e_ℓ* is ℓ*-th standard basis vector`

**9**    $\tilde{\mathbf{X}} \leftarrow \begin{bmatrix} \mathbf{c}_1^{(k)}\mathbf{X}_1/\|\mathbf{X}_1\| \in \mathbb{R}_{\geq 0}^{m_1 \times n} \\ \vdots \\ \mathbf{c}_L^{(k)}\mathbf{X}_L/\|\mathbf{X}_L\| \in \mathbb{R}_{\geq 0}^{m_L \times n} \end{bmatrix}, \tilde{\mathbf{W}} \leftarrow \begin{bmatrix} \mathbf{c}_1^{(k)}\mathbf{W}_1^{(k-1)}/\|\mathbf{X}_1\| \in \mathbb{R}_{\geq 0}^{m_1 \times r} \\ \vdots \\ \mathbf{c}_L^{(k)}\mathbf{W}_L^{(k-1)}/\|\mathbf{X}_L\| \in \mathbb{R}_{\geq 0}^{m_L \times r} \end{bmatrix}$

**10**    $\mathbf{H}^{(k)} \leftarrow \mathbf{H}^{(k-1)} \odot \dfrac{\tilde{\mathbf{W}}^\top \tilde{\mathbf{X}}}{\tilde{\mathbf{W}}^\top \tilde{\mathbf{W}} \mathbf{H}^{(k-1)}}$

**11**    $\mathbf{W}^{(k)} = \begin{bmatrix} \mathbf{W}_1^{(k)} \\ \vdots \\ \mathbf{W}_L^{(k)} \end{bmatrix} \leftarrow \mathbf{W}^{(k-1)} \odot \dfrac{\mathbf{X}\mathbf{H}^{(k)\top}}{\mathbf{W}^{(k-1)}\mathbf{H}^{(k)}\mathbf{H}^{(k)\top}}$

**Output:** coefficient matrix $\mathbf{W}^{(k)} = \begin{bmatrix} \mathbf{W}_1^{(k)} \in \mathbb{R}_{\geq 0}^{m_1 \times r} \\ \vdots \\ \mathbf{W}_L^{(k)} \in \mathbb{R}_{\geq 0}^{m_L \times r} \end{bmatrix}$; dictionary matrix $\mathbf{H}^{(k)} \in \mathbb{R}_{\geq 0}^{r \times n}$

## 6 Numerical Experiments

In this section, we perform numerical experiments on a synthetic dataset used as a benchmark dataset and two real datasets representing data types or scenarios where NMF is typically used. For the synthetic dataset, we consider groups with different sizes and complexities. The first real dataset is the Heart Disease dataset (Janosi et al., 1989) with two groups "female" and "male". The second is the 20Newsgroups dataset (Rennie, 1995), a common benchmark dataset used in document classification, clustering, and topic modeling.

In the numerical experiments discussion, we will use the acronym R-Error for the average relative reconstruction error and R-Loss for the average relative reconstruction loss.

### 6.1 Data Pre-processing and Implementations

In all of the datasets, we normalize the features to have unit $\ell_2$-norm. We estimate $E_\ell$ defined in Remark 4.2 as Algorithm 1 with 5 runs of NMF ($T = 5$).

We evaluate the performance of the AM scheme (Algorithm 2) and the MU scheme (Algorithm 3) for Fairer-NMF. For the AM experiments, we use the open-source package CVXPY (Diamond & Boyd, 2016), which supports a number of different specialized convex program solvers. The specific solvers we use are ECOS (Domahidi et al., 2013) and SCS (O'Donoghue et al., 2016) for the SOCP to find $\mathbf{H}$ and OSQP (Stellato et al., 2020) for the QP to find $\mathbf{W}$. For the SOCP, we default to ECOS and switch to SCS only if ECOS fails. Failures of ECOS do occur, but only for large problem sizes.

For all the plots, we report the mean and standard deviation (given by the shaded region around the mean) over 10 trials. We report the "Relative Error (%)" to be the relative reconstruction error of NMF given in Definition 3.1 scaled by 100. For both methods, we iterate until the change in each group's reconstruction error in a single iteration is no more than $10^{-4}$ times the current reconstruction error. That is when the following condition is met:

$$\max_{\ell \in \{1,\ldots,L\}} \frac{\left| \|\mathbf{X}_\ell - \mathbf{W}_\ell^{(k)}\mathbf{H}^{(k)}\| - \|\mathbf{X}_\ell - \mathbf{W}_\ell^{(k-1)}\mathbf{H}^{(k-1)}\| \right|}{\|\mathbf{X}_\ell - \mathbf{W}_\ell^{(k)}\mathbf{H}^{(k)}\|} < 10^{-4}. \tag{10}$$

### 6.2 Synthetic Dataset

We first consider a synthetic dataset consisting of groups of varying sizes and ranks. We generate the data matrix $\mathbf{X} \in \mathbb{R}_{\geq 0}^{m \times n}$ as $\mathbf{X} = \mathbf{WH}$ where $\mathbf{W} \in \mathbb{R}_{\geq 0}^{m \times r}$ and $\mathbf{H} \in \mathbb{R}_{\geq 0}^{r \times n}$ are sampled from a uniform distribution over $[0, 1)$. We consider three different combinations of sizes and ranks to generate the groups as given by Table 1.

Table 1: Synthetic data parameters

| Name | $m$ | $n$ | $r$ |
|------|-----|-----|-----|
| Large size low rank | 1000 | 20 | 3 |
| Medium size low rank | 500 | 20 | 3 |
| Small size high rank | 250 | 20 | 6 |

As discussed in Section 4.1, Figure 2 shows that Standard NMF exhibits a discrepancy in the R-Error among the three groups. In Figure 3, we observe the small-size group incurring the highest loss for ranks 6 to 11. Here, the loss can be interpreted as the difference between the reconstruction error the group would incur by being part of the population and the error the group would have incurred if the model was run on that group alone. Thus, Figure 3 confirms that the small-sized group is "sacrificing" the most (for ranks 6 and higher) by being part of the population. This is of course not surprising, given that NMF minimizes error "on average" and is thus likely to ignore smaller groups. In Figure 4 for Fairer-NMF, we see how the high-rank

group incurs the largest R-Error for ranks 3, 4, and 5 and that all groups have a comparable R-Loss (Figure 5). Indeed, Fairer-NMF seeks to minimize the maximum relative reconstruction loss.

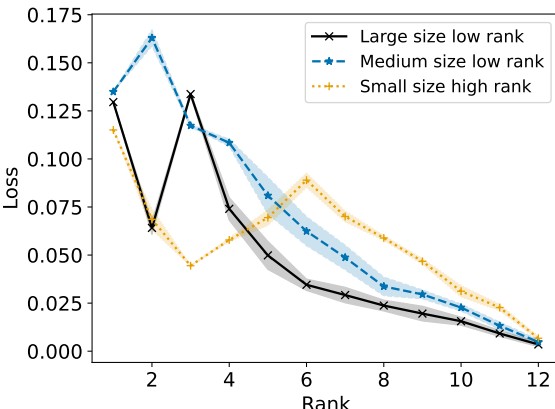

Figure 3: Standard NMF applied on the entire synthetic data matrix. The reconstruction loss of each group is reported.

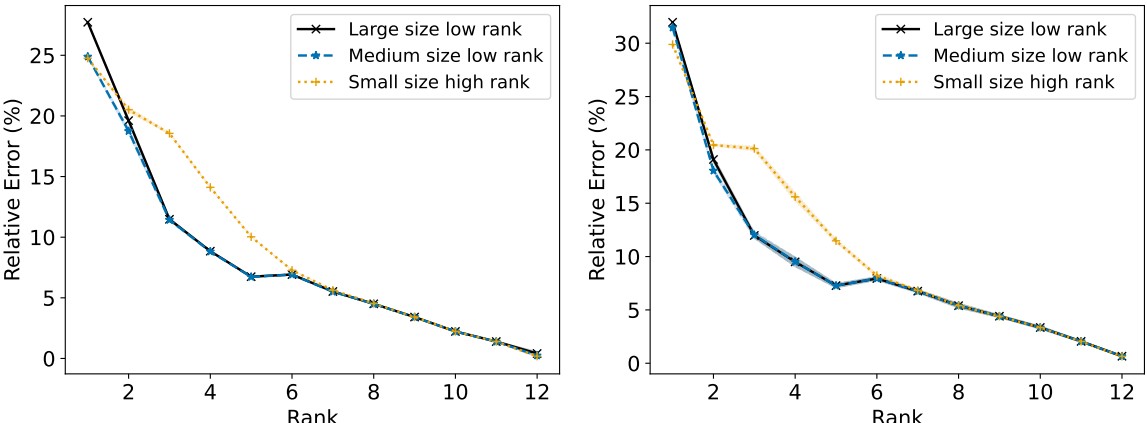

Figure 4: Fairer-NMF applied on the synthetic data matrix. The Relative Error (%) of each group is reported. Left: Fairer-NMF with the alternating minimization scheme. Right: Fairer-NMF with the multiplicative updates scheme.

### 6.3 Heart Disease Dataset

The heart disease dataset (Janosi et al., 1989) is a dataset designed for medical research to predict whether a patient has heart disease or not based on various medical attributes. The dataset is commonly used in machine learning research to evaluate a model's performance in classifying the presence and absence of the disease. The most complete and commonly used subset of the dataset in machine learning research is the Cleveland database. The database consists of 303 samples and 13 attributes that are clinical parameters obtained from the patients such as sex, age, resting blood pressure, and serum cholesterol.

In our experiments, we seek to investigate the performance of NMF in representing the population stratified by patient sex (reported only as male or female in this dataset). We conduct this analysis in an unsupervised setting, excluding the binary target variable that indicates disease presence or absence. We also omit the sex attribute to perform our analysis on the two populations. The numerical features in the dataset are

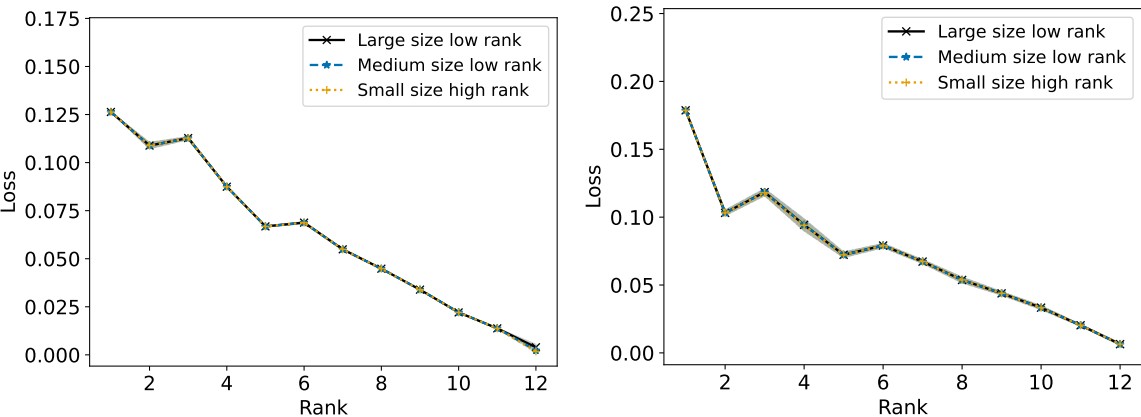

Figure 5: Fairer-NMF applied on the synthetic data matrix. The reconstruction loss of each group in the dataset is reported. Left: Fairer-NMF with the alternating minimization scheme. Left: Fairer-NMF with the multiplicative updates scheme.

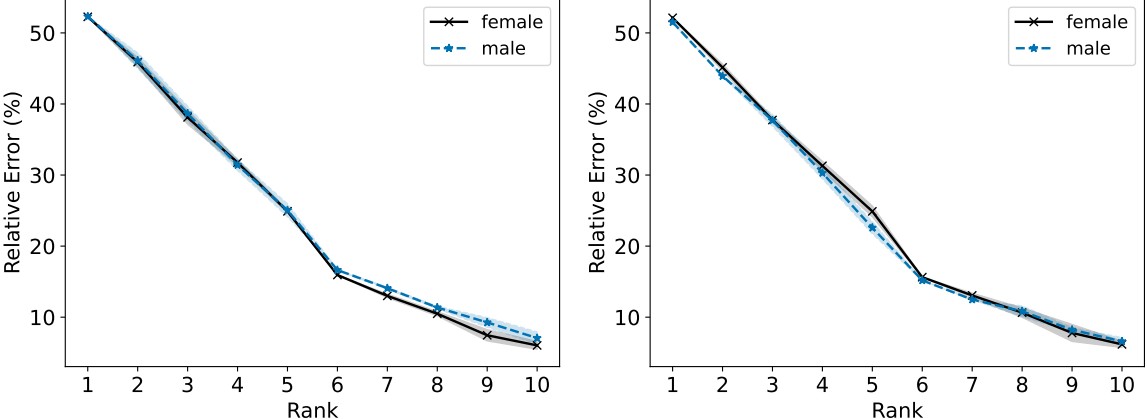

Figure 6: The Relative Error (%) of each group in the Heart Disease dataset is reported. Left: Standard NMF applied on the entire Heart Disease data matrix. Right: Standard NMF applied on the Heart Disease data matrix of each group individually.

non-negative and the categorical features are recorded as integers. There are 201 individuals in the female group and 96 in the male group.

In the right plot of Figure 6, we observe that the male population generally incurs a slightly lower R-Error than the female population for the same low-rank NMF models. In Figure 7, we observe a generally higher R-Loss for the male group compared to the female group which indicates that Standard NMF inadvertently favored the female group.

As observed in Figure 9, generally Fairer-NMF achieves a similar loss for both populations which in this application may or may not be "fair". With the fairness criterion considered in Fairer-NMF, some patients will incur a higher reconstruction error compared to that achieved with a Standard NMF model.

We highlight a counterintuitive phenomenon that appears in Figures 7 and 9. In both of these plots, the loss is negative at some of the ranks for at least one of the groups. Even though we consider the "excess reconstruction error" $\|\mathbf{X}_\ell - \mathbf{W}_\ell \mathbf{H}\| - E_\ell$ as part of our loss, this quantity need not be positive. Since we only choose $E_\ell$ to be the expected error for a single group NMF, a reconstruction may obtain better error than $E_\ell$ for a single group. This leaves room for the loss to be negative not only due to random chance but also due to the existence of a reconstruction that standard NMF on the individual group was unable to find. While

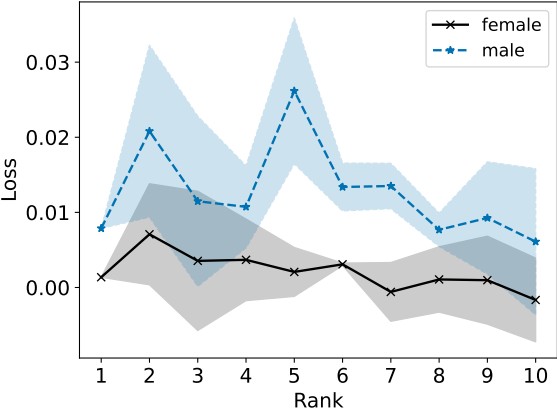

Figure 7: Standard NMF applied on the entire Heart Disease data matrix. The reconstruction loss of each group is reported.

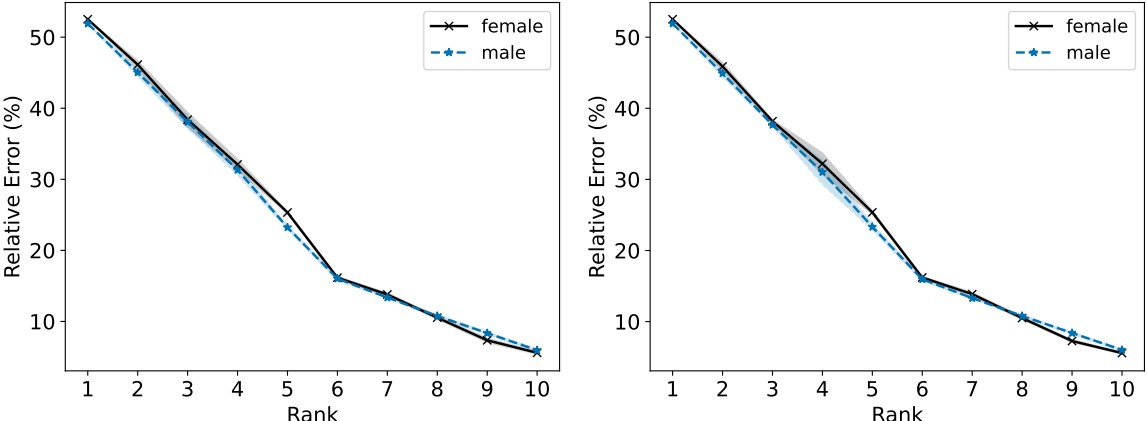

Figure 8: Fairer-NMF applied on the Heart Disease data matrix. The Relative Error (%) of each group is reported. Left: Fairer-NMF with the alternating minimization scheme). Left: Fairer-NMF with the multiplicative updates scheme.

we might expect that adding additional groups would make it less likely to find a good reconstruction for the $\ell$-th group, this is not always the case. Indeed, in Figure 9 we see that the loss for a rank-10 Fairer-NMF is consistently below 0, meaning that Fairer-NMF is able to find a better reconstruction for each group when applied to the full dataset than standard NMF can even when applied to just a single group.

### 6.4 20Newsgroups Dataset

The 20newsgroups dataset (Rennie, 1995) is a popular benchmark dataset containing documents gathered from 20 newsgroups that are partitioned into 6 major subjects. We sample 1500 documents from the entire dataset with the number of samples from each subject is proportional to the size of the subject in the entire dataset. We cast all letters to lowercase and remove special characters as part of the pre-processing of the data. The TFIDF vectorizer with the English stop words list from the NLTK package is applied to transform the text data into a matrix. After obtaining the data matrix of the entire dataset, the matrix is partitioned into 6 groups according to the 6 subjects present in the original dataset. The sizes of the groups are: Computer 389, Sale 78, Recreation 316, Politics 209, Religion 193, Scientific 315.

Figure 10 shows the R-Error of each group with standard NMF, both applied to the full dataset at once and each group individually. When NMF is performed on each group individually, the R-Error at rank 20 ranges

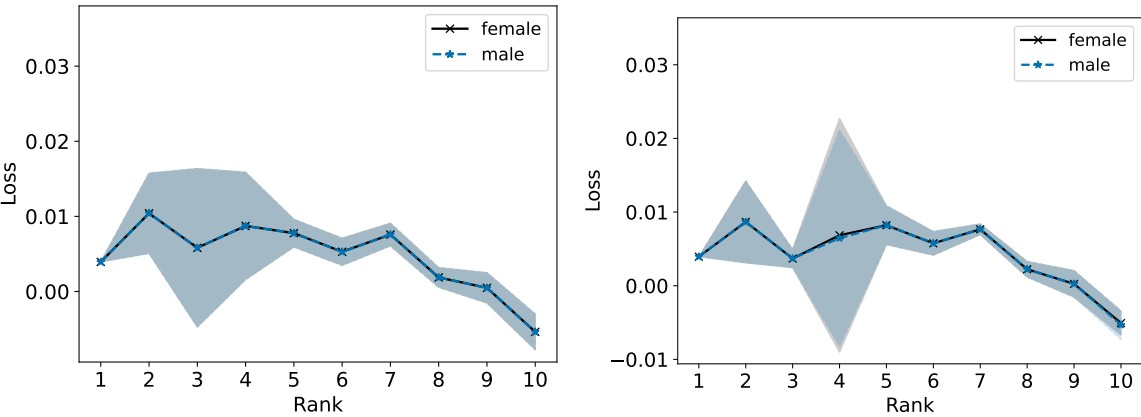

Figure 9: Fairer-NMF applied on the Heart Disease data matrix. The reconstruction loss of each group in the dataset is reported. Left: Fairer-NMF with the alternating minimization scheme. Left: Fairer-NMF with the multiplicative updates scheme.

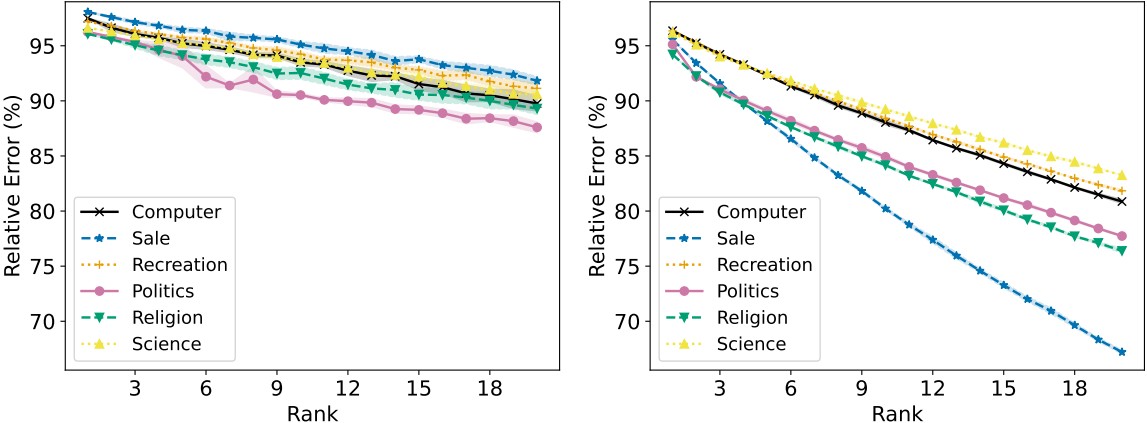

Figure 10: The Relative Error (%) of each group in the 20 Newsgroups dataset is reported. Left: Standard NMF applied on the entire 20 Newsgroups data matrix. Right: Standard NMF applied on the 20 Newsgroups data matrix of each group individually.

from around 70% to 85%, depending on the group. When one NMF reconstruction is obtained for the full dataset, the range of R-Error is more tightly clustered around 90%. Notably, the "Sale" group has the lowest R-Error when a standard NMF construction is obtained for each group individually, but the highest when one is obtained from all groups at once. This results in it having the highest R-Loss, as shown in Figure 11.

Fairer-NMF is able to resolve this discrepancy. Figure 12 shows that the six different groups have reconstruction errors that correspond to the individual group reconstruction errors. The "Sale" group, being the easiest to explain with a low-rank approximation, is the group with the lowest R-Error, and the rest of the groups appear in the order they do in the right plot of Figure 10. Accordingly, the loss of each group under Fairer-NMF, as shown in Figure 13, are all very similar.

Figures 12 and 13 show another counterintuitive phenomenon. While R-Error for each group decreases as the number of ranks in the decomposition increases, R-Loss increases as the rank goes up. This is because the reconstruction error of each group decreases in rank much faster when decomposed individually than when decomposed together.

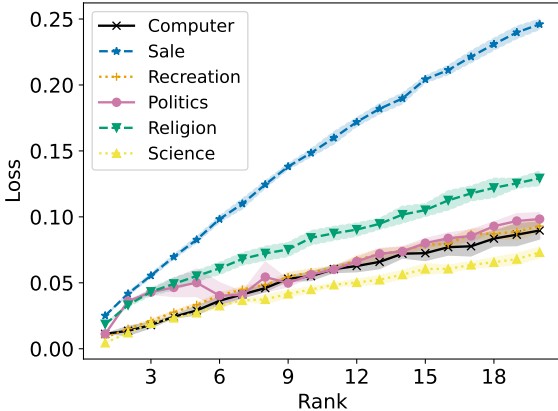

Figure 11: Standard NMF applied on the entire 20 Newsgroups data matrix. The reconstruction loss of each group is reported.

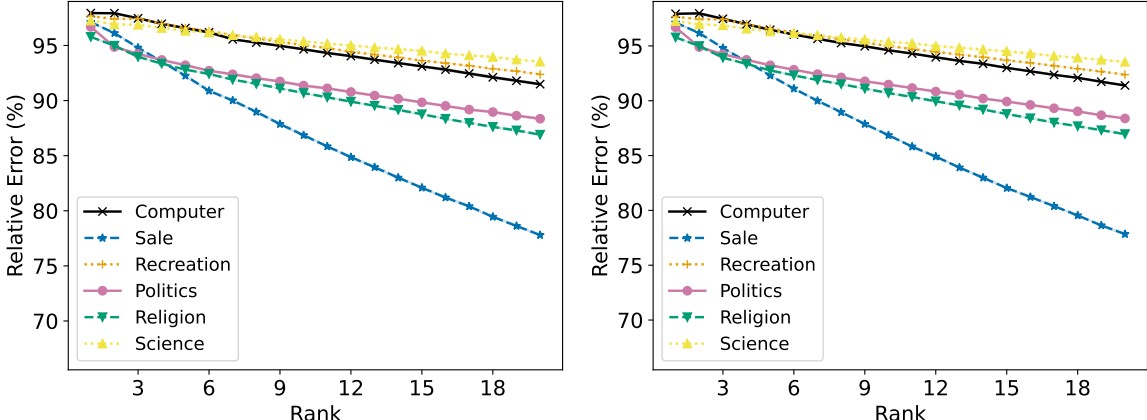

Figure 12: Fairer-NMF applied on the 20 Newsgroups data matrix. The reconstruction loss of each group in the dataset is reported. Left: Fairer-NMF with the alternating minimization scheme. Left: Fairer-NMF with the multiplicative updates scheme.

## 6.5 Algorithm Comparisons

We propose two different algorithms for Fairer-NMF: the alternating minimization method (Algorithm 2) and the multiplicative update method (Algorithm 3). In Figures 5, 9, and 13, both algorithms are run on three different datasets: the synthetic dataset, the heart disease dataset, and the 20Newsgroups dataset, respectively. For the heart disease and 20Newsgroups datasets, both algorithms perform equally well. However, for the synthetic dataset, the alternating minimization method is able to consistently find a lower loss solution for Fairer-NMF than the multiplicative update method. This discrepancy is mirrored in Section 5, where we only show a non-increasing result for the alternating minimization method.

However, an important consideration when choosing between these two methods is the computation cost of each. The alternating minimization involves solving one SOCP and one NNLS problem with each iteration, which is very expensive. On the other hand, the multiplicative update scheme only requires a few matrix multiplications in each iteration. Figure 14 shows how long both algorithms took to reach convergence on dataset and rank combination when run on a 12 core 3.50GHz Intel i9-9920X CPU. For the larger datasets (the 20Newsgroups dataset and higher ranks of the synthetic dataset), the alternating minimization problem is substantially slower than the multiplicative update. A single Fairer-NMF decomposition with

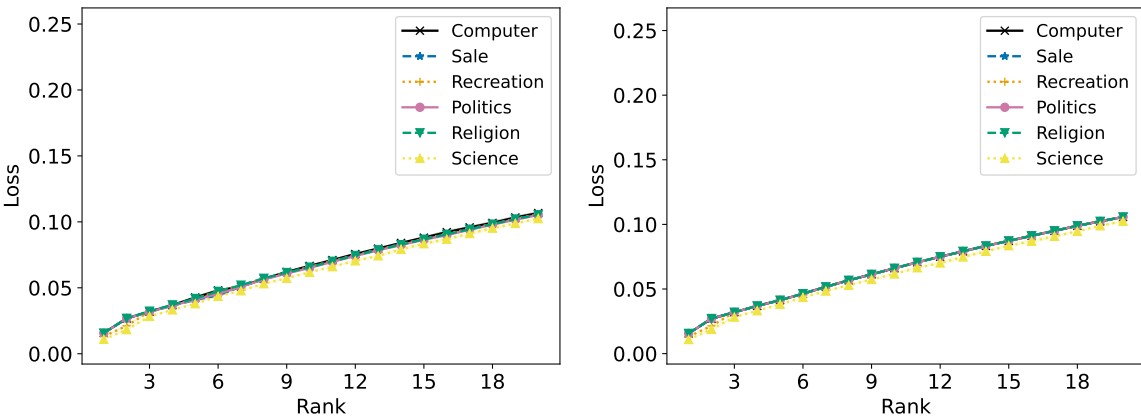

Figure 13: Fairer-NMF applied on the 20 Newsgroups data matrix. The reconstruction loss of each group in the dataset is reported. Left: Fairer-NMF with the alternating minimization scheme. Left: Fairer-NMF with the multiplicative updates scheme.

the alternating minimization method can easily take over an hour to perform, whereas the longest time for convergence with the multiplicative update method across all datasets and ranks is 129 seconds.

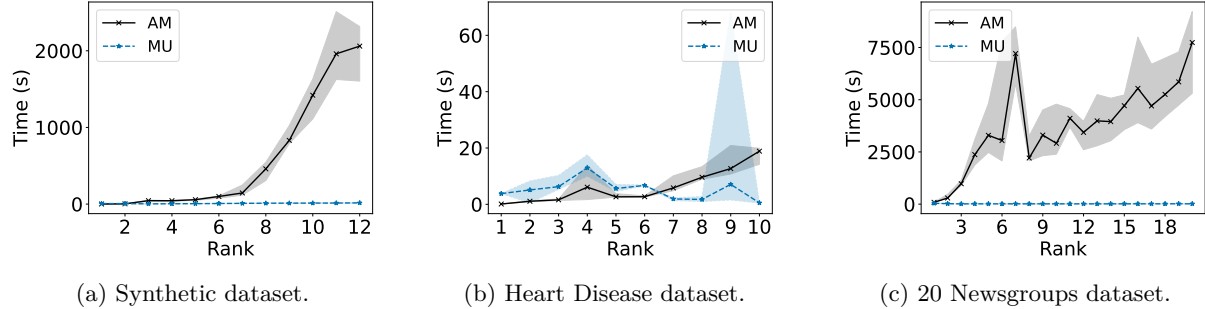

(a) Synthetic dataset.  (b) Heart Disease dataset.  (c) 20 Newsgroups dataset.

Figure 14: Time in seconds required for Fairer-NMF with alternating minimization (AM) scheme and with multiplicative updates (MU) scheme to converge for all three datasets. The convergence criterion is given by (10). We report the median and interquartile range over 10 trials.

## 7 Discussion

We remark here on the title of the manuscript, and the notion that our proposed framework, and indeed any machine learning framework, is very unlikely to ever be completely "fair". On the other hand, there is certainly a need to make ML algorithms *fairer*—to help practitioners identify inequities and provide alternative methods that offer a fairer outcome for some applications. Our objective in (5), for example, asks that the maximum reconstruction loss across all population groups be minimized. In many contexts, as motivated in Section 4, this results in fairer outcomes. Indeed, in many settings, populations consist of majority groups and minority groups, and because typical models minimize average or overall error, minority groups will typically have a higher reconstruction error than the majority. Further, Standard NMF does not take into account the complexity of the groups.

The objective we propose takes into account the size and complexity of the groups and achieves fairer outcomes in these settings. However, some drawbacks need to be considered, as will be the case for any method that attempts to mitigate fairness. First, we assume the population groups are known apriori. This can likely be overcome by learning the groups on the fly through cross-validation of reconstruction errors and is a future direction of research. The next concern of course is that it may not always be desirable

to minimize the maximum reconstruction loss. Indeed, through this fairness mitigation, some groups and therefore individuals may receive a higher reconstruction error than they would have without the "fairer" approach. In settings like medical applications, where these tools are used to predict, for example, the likelihood of a patient having a disease, this may no longer seem fairer. It is thus clear that fairness mitigation is highly application-dependent. The same is true of group size and complexity. As some of our experiments show, a group that is complex to explain (e.g., has a high rank on its own), may dominate the factorization when using our proposed objective. In extreme cases, this may not be desirable, since it may prevent other, less complex groups from receiving good reconstruction error. This is all to say that, the notion of fairness itself is highly application-dependent, and great care should be taken when mitigating—or not—in learning methods.

## 8 Conclusion

NMF is a widely used topic modeling technique in various domains, particularly when interpretability and trust are essential. We believe that examining the fairness of NMF is a valuable contribution to the field and an important step toward tackling key issues related to bias and fairness. In this work, we presented an alternative NMF objective that seeks a non-negative low-rank model that provides equitable reconstruction loss across different groups. The goal is to learn a common NMF model for all groups under the min-max fairness framework which seeks to minimize the maximum of the average reconstruction loss across groups. We proposed an alternating minimization algorithm and a multiplicative updates algorithm. Numerically, the latter demonstrated reduced computational time compared to a CVXPY (Diamond & Boyd, 2016) implementation of the AM algorithm while still achieving similar performance. We showcased on synthetic and real datasets how standard NMF could lead to biased outcomes and discussed the overall performance of Fairer-NMF.

## Acknowledgments

EG is partially supported by the UCLA Racial Justice seed grant, UCLA Dissertation Year Award, and the NSF Graduate Research Fellowship under grant DGE 2034835. LK and DN are partially supported by the Dunn Family Endowed Chair fund. DN is partially supported by NSF DMS 2408912. The authors EG and DN worked on this material while in residence at the Simons Laufer Mathematical Sciences Institute (formerly MSRI) in Berkeley, California, during the Fall 2023 semester.

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
