# OpenReview forum: "Towards a Fairer Non-negative Matrix Factorization"
_TMLR — Rejected by TMLR_

### Review · Reviewer_jBop · 2024-12-11

**Summary Of Contributions:**

This paper proposes **fairer** NMF. A minor adaptation of classical NMF to account for group size while minimizing the maximum reconstruction loss. Both, Alternative Minimization and Multiplicative Update are adapted to this changed objective:
$\mathcal{L} = \sum_l \frac{\left(||X_l - WH|| - E_l\right)}{||X_l||}, \quad \text{where } E_l = \mathbb{E}\left(x^*_l \cdot ||X_l - X^*_l||\right)$

**Audience:**

No

**Claims And Evidence:**

No

**Requested Changes:**

As described in the weaknesses section, I do not believe that the method is a suitable fit for TMLR, largely independent of any changes. The only way I currently see for this manuscript to move forward would be through a comprehensive benchmark study demonstrating the effectiveness of the proposed method. Even then, I remain somewhat skeptical, given that the method represents only a minor adaptation of existing approaches.

**Strengths And Weaknesses:**

**Strengths**
- The paper is overall well-written, with concise and clear notation.
- The idea and problem set are well-introduced.

**Weaknesses**
- Introducing fair PCA and FGLM first and then (re)-introducing the notation feels redundant.
- Large portions of the paper are dedicated to detailed explanations of well-known methods or comparable algorithms. Nearly four full pages on fair PCA, FGLM, and NMF are excessive. This would be more appropriate for an introductory paper on these methods, but in my opinion, not when introducing a new method.
    - The figure (Figure 1) in the NMF section appears arbitrary.
- While both fair PCA and FGLM are introduced, they are not compared to the proposed fairer NMF.
- The experiments lack both size and detail.
   - The plots comparing NMF and fairer NMF are not directly comparable due to different axis values.
   - Relying solely on plots for analysis and drawing strong conclusions from a limited number of experiments is an overstatement.

**General Weakness**

If I am not mistaken, the only contribution of the proposed method is the adapted loss function, with a minor modification to classical NMF. Even the AM and MU algorithms are nearly identical, with changes limited to this loss function.
Overall, I am not convinced that these minor adaptations are a suitable fit for TMLR, particularly given the lack of larger experiments or real-world applications. I am open to being convinced otherwise by fellow reviewers, authors, or ACs, but in its current state, I do not believe this paper is suitable for publication in TMLR.

---

> ### Author Response · Authors · 2024-12-17
> **Response and actions**
>
> We thank the reviewer for their constructive feedback, which helped us identify areas to improve the clarity and impact of our work. Below, we address each point raised:
>
> 1. We will address the reviewer's concern about redundancy and length by reducing repetitive content and sections on existing approaches.
>
> 2. We will ensure that the y-axis range is consistent across the figures.
>
> 3. We will conduct larger-scale and more comprehensive experiments. Specifically, we will expand the synthetic datasets to include a larger range of size-rank combinations and structures in the data, such as orthogonal bases. We will also include an additional real-world dataset in our experiments to further demonstrate the practical effectiveness and applications of Fairer-NMF.
>
> 4. We do disagree with the reviewer's assessment that our contribution is a minor modification from NMF. Unlike the standard NMF objective function, Fairer-NMF takes into account the complexity of data groups, a major and critical factor to consider while studying unintentional bias introduced by dimension reduction techniques such as NMF. This is motivated from Fair PCA, but in the case of PCA, the update derivation and analysis is much more straightforward because of orthogonal cancellations that NMF does not benefit from. We will revise the manuscript to more clearly emphasize and articulate the importance of this contribution.
>
> While we don't disagree that our result is not a deep theoretical analysis, we believe it is an important and timely contribution to the ML literature. However, if the reviewers and editors believe it to be a better fit elsewhere than TMLR we will withdraw and are grateful for the helpful suggestions.

---

### Review · Reviewer_rd6w · 2025-01-08

**Summary Of Contributions:**

The authors study how representations learned via nonnegative matrix factorization can introduce bias in different subgroups of the data, e.g., as defined by gender, race, or other protected attributes. They measure fairness according to the maximum reconstruction error across different subgroups (relative to subgroup size and intrinsic complexity) and propose two algorithms for minimizing this metric. Experiments on synthetic and real datasets demonstrate the benefits of the proposed methods.

**Audience:**

Yes

**Claims And Evidence:**

No

**Requested Changes:**

Please address each of the points mentioned in the weaknesses section.

This is minor, but the citations in the paper do not have links to the corresponding papers in the References section. It is not required, but it would be a quality of life improvement for readers if these links were included, so that the interested reader can click on the in-text citation and be taken to the corresponding full citation in the References section. This can be accomplished with the natbib LaTeX package, among other options. The same is true for figure and definition references; adding links to these would make the paper much easier to navigate.

Typos:
1. "multiplicative scheme (MU) scheme" (Middle of the paragraph in Section 1.1).
2. The index k should be incremented in the AM description on pg. 5.

**Strengths And Weaknesses:**

# Strengths

The paper is well-written and easy to read.

As the authors discussed, fairness concerns are of growing importance as ML methods are applied in more socially relevant settings. Thus, the topic studied by the paper should be relevant to the TMLR audience.

Explanations of some of the more counterintuitive results/plots were provided which makes it easier to grok the paper. For instance, the discussion of why the R-Loss *increases* with rank at the bottom of pg. 16 was helpful, as was the explanation for why the multiplicative update scheme is self-correcting on pg. 11.

The discussion of limitations and nuances involved with any definition of fairness are thorough. In spite of the difficulties of proposing a universally acceptable notion of fairness, the notion that the authors use does seem reasonable, and I agree that it does capture notions of intrinsic difficult of different subgroups.

The proposed algorithms are intuitive and seem to achieve the desired result on the datasets analyzed. (Slight caveat on the synthetic data in the weaknesses section below, where the relative performance of standard vs. fair NMF is unclear.)

# Weaknesses

While the general topic of fairness is certainly relevant to the TMLR audience, the specific application to NMF may be of limited interest. In particular, I'm not sure that the claim that "NMF has garnered increasing attention" is true in modern ML, and most of the papers cited by the authors as applications of NMF are from the early 2000s or late 1990s.

I was not certain of the definition of the synthetic dataset. My assumption was that there is a *different* H matrix which is randomly sampled for each group. Then, each group is generated according to X_l = W_l H_l, and finally the data are combined into a single data matrix. Is this correct?

There is no direct comparison of Standard NMF and Fairer-NMF on the synthetic data. Fig. 3 reports the R-Loss for Standard NMF, while Fig. 4 reports the R-Error for Fairer-NMF. In particular, it's not possible to tell just from these tables whether or not Fairer-NMF has made the error for each group worse than it was with Standard NMF, which seems clearly undesirable even if the errors for each group have been equalized.

More generally, even though comparable metrics are applied to the two methods (standard and fairer NMF) for the real datasets, the results are given in separate plots. These plots should be combined so that it is easier to get a sense of the relative performance of the fair vs. standard NMF.

There is no discussion of Fig. 5.

Acknowledgements should be removed for the blind submission.

---

> ### Author Response · Authors · 2025-01-14
> **Response and actions**
>
> We thank the reviewer for their helpful comments and suggestions. Below, we address each point raised:
>
> 1. We believe that NMF is both a useful method in its own right and also an example of an ML technique that can benefit from fairness analysis, more complicated than that of PCA. We will include more modern references including some recent surveys that highlight a number of current applications.
> 2. We will clarify the creation of the synthetic dataset, include more comparison plots (which we initially did not include due to space), and discussion of the experiments.
> 3. Third Paragraph of the Weaknesses Section: Figures 2-5 report the results of the synthetic experiment. Figure 2 reports the R-Error for Standard NMF as referenced in the second paragraph of Section 6.2.
> 4. Fourth Paragraph of the Weakness Section: Thank you for your suggestion. We have some concerns that combining the NMF and Fairer-NMF plots might result in a visualization that is too complex, particularly for the 20 Newsgroups dataset. We worry this could make the results harder to interpret.
> 5. We will adjust the citations to have links and remove acknowledgments.
> 6. Thank you for pointing out the typos. We will fix these.

---

### Review · Reviewer_itV3 · 2025-01-31

**Summary Of Contributions:**

The paper addresses group fairness for non-negative matrix factorization where the matrix rows are partitioned into L groups. An objective is formulated in terms of worst-case reconstruction loss over groups, where the loss compares the overall reconstruction error to the error that would be obtained by factorizing the group independently. The classical alternative-minimization and multiplicative-update algorithms are modified to minimize this objective. Numerical experiments show that this results in each group having almost the same loss. The multiplicative update method is shown to be more computationally scalable.

**Audience:**

Yes

**Broader Impact Concerns:**

no special concerns

**Claims And Evidence:**

Yes

**Requested Changes:**

As noted above, please formalize the discussion of the self-correcting property mentioned in 5.3. If possible, characterize the sensitivity to E_l, which is estimated by a randomized algorithm. For example, how many iterations are necessary in order to be confident of the reconstruction errors produced by the NMF algorithm?

From the figures, it is very hard to compare the per-group errors/losses with and without the fairness objective. For example, between figs 5-9 it would be good to have one figure showing the per-group losses with standard and fair NMF. Similarly for 20 newsgroups. This would illustrate the fairness tradeoff for groups that are well- and poorly-reconstructed in the original NMF.

Minor:

- In definition 4.1, I think $W$ should be $W_l$ or the dimensions don't match
- On page 10, the function $g$ is not defined

**Strengths And Weaknesses:**

Overall this is a clearly written paper with a solid solution to a well-stated problem. The experiments support the basic intuitions offered in the derivations.

I am not an expert in this area and so I cannot be sure about the novelty, but I am not aware of prior work that offers the same contributions.

From a theoretical perspective, I would appreciate more formality in the discussion of the apparently self-correcting property of the algorithm in section 5.3.

I would also like to know more about the irreducible error term E_l, and how sensitive are the results to the computation of this term.

---

> ### Author Response · Authors · 2025-02-11
> **Response and actions**
>
> We thank the reviewer for their valuable feedback on how to make the paper more understandable. Below, we address each point raised:
> 1. Yes, it should be $W_\ell \in \mathbb{R}^{m_\ell \times r}$ in Definition 4.1. Thank you for catching this typo.
> 2. Thank you for the comment. We will be more explicit in defining the function $g$.
> 3. We will clarify the self-correcting property.  We do not have a formal derivation. Therefore, we will also make it clear that it is conjectured.
> 4. The sensitivity to $E_\ell$ is revealed in our experiments.  We plot the standard deviation of the measures as a shaded region, where the standard deviation captures the random initialization of Fairer-NMF and the randomness of estimating $E_\ell$.  We will clarify this in the paper and call attention to this demonstrating only mild sensitivity to $E_\ell$.
> 5. Thank you for the comment and suggestion regarding the ease of comparisons among the methods in the figures. We will have a figure consisting of three subplots side by side with the same y-axis for the reconstruction error of NMF, Fairer-NMF (AM), and Fairer-NMF (MU). We will also have a figure of the individual group reconstruction errors with the same y-axis as the reconstruction error subplots. We will similarly have a figure consisting of three subplots side by side with the same y-axis for the reconstruction loss of NMF, Fairer-NMF (AM), and Fairer-NMF (MU).

---

### Author Response · Authors · 2025-02-11
**Response and actions**

We again thank all the reviewers for their constructive feedback.  In response, we intend to make the following changes promised in the response to the reviews.  In summary,

1. We will add more experimental results on synthetic data and the Labeled Faces in the Wild dataset.
2. We will update the figures so that they can be compared more easily to each other. In particular, we will make sure the $y$-axis scales are consistent across plots on the same data and will show the Fairer-NMF plots side by side with standard NMF.
3. We will add more references and streamline our discussion of previous work.
4. We will clarify the discussion of our methods and results as mentioned in each individual reviewer response.

---

### Author Response · Authors · 2025-04-01
**Status inquiry**

Hello, we would like to inquire about the status of our manuscript and if there are any other steps we need to take. Thank you!

---

> ### Comment · Action_Editor_5f9e · 2025-04-02
> **decision made**
>
> The AE recommendation has been posted.

---

### Decision · Action_Editor_5f9e · 2025-04-02

**Recommendation:** Reject

**Comment:**

This paper presents an extension of non-negative matrix factorization (NMF) to ensure fairness in group data settings. The proposed method "faired NMF" is based on minimization of a new loss function for NMF, which is basically defined as the maximum reconstruction loss across different groups (normalized by group sizes and complexities). Notably, such an objective is inspired by prior work on fair PCA (FPCA)  and fair generalized low-rank models (FGLM) of which fair NMF would be a special case, both of which also use such min-max formulations.

The paper received 3 reviews with 2 reviewers leaning towards rejection and one reviewer leaning towards acceptance.

The reviewers leaning towards rejection expressed several concerns, in particular: (1) the proposed method being a minor adaptation (of the loss function) of standard NMF, especially given works on FPCA and FGLM; (2) lack of a comprehensive empirical evaluation on more datasets; (3) much of the paper describing background of standard techniques, and not adequate analysis and experiments to study the effectiveness of fairer NMF on real-world tasks.

The paper also lacks a theoretical analysis of the proposed method unlike prior works such as fair PCA. While this is okay, in such a case, the paper should have been stronger in terms of experimental evaluation.

The reviewers  made several suggestions to improve the manuscript. However, while the authors promised to make the suggested changes, no revised version of the paper was submitted. In the absence of a revised version, it was not possible for the reviewers to reconsider their original assessment. Also, in the discussions with reviewers, no reviewer championed the paper, and the AE agrees with the reviewers' view that the paper in its current form is not strong enough for acceptance at this venue.

We suggest the authors to consider incorporating the feedback from the reviewers and consider resubmitting to another venue.

**Audience:**

Yes, this paper is on fair NMF and it may be of interest to researchers and practitioners interested in interpretable unsupervised data analysis (for which NMF is still considered one of the classic and trusted methods) and group fairness.

**Claims And Evidence:**

The paper was evaluated in terms of its claimed contributions, i.e., the empirical benefits of the proposed fair NMF method for fair representation learning.

However, as the reviewers have also pointed out, the empirical evidence in the paper is fairly limited. Given that there is no theoretical analyses of the proposed method (which is fine), and the paper had to be evaluated in terms of the empirical benefits of the proposed fair NMF over standard NMF (and ideally over other methods too, such as fair PCA and fair GLM), the empirical evaluation here should have been stronger.

Even if we ignore the lack of comparison with methods such as fair PCA or fair GLM (one the the special cases of the latter is actually fair NMF),  the comparisons with standard NMF are also not comprehensive and rigorous, and are on rather modest-scale datasets. The reviewers too raised the issue of inadequate empirical evaluation. There were also several issues with the presentation of the results.

The authors did acknowledge these issues in the discussion but  didn't submit a revised manuscript to address those issues.